# Potential Use of Papaya Waste as a Fuel for Bioelectricity Generation

**Segundo Rojas-Flores [1],\*** , **Orlando Pérez-Delgado [1]** , **Renny Nazario-Naveda [2]** , **Henry Rojales-Alfaro [3]** ,
**Santiago M. Benites [4]** , **Magaly De La Cruz-Noriega [4]** and **Nélida Milly Otiniano [4]**

1   Research Laboratory of Health Sciences, Universidad Señor de Sipan, Chiclayo 14000, Peru;
    comoperezd@gmail.com
2   Research Group in Applied Sciences and New Technologies, Universidad Privada del Norte,
    Trujillo 13007, Peru; renny.nazario@upn.edu.pe
3   Estudios Generales, Universidad Privada del Norte, Trujillo 13007, Peru; henry.rojales@upn.edu.pe
4   Instituto de Investigación en Ciencias y Tecnología, Universidad Cesar Vallejo, Trujillo 13001, Peru;
    sbenites@ucv.edu.pe (S.M.B.); mdelacruzn@ucv.edu.pe (M.D.L.C.-N.); notiniano@ucv.edu.pe (N.M.O.)
\*   Correspondence: segundo.rojas.89@gmail.com

**Abstract:** Papaya (*Carica papaya*) waste cause significant commercial and environmental damage,
mainly due to the economic losses and foul odours they emit when decomposing. Therefore, this work
provides an innovative way to generate electricity for the benefit of society and companies dedicated
to the import and export of this fruit. Microbial fuel cells are a technology that allows electricity
generation. These cells were produced with low-cost materials using zinc and copper electrodes;
while a 150 mL polymethylmethacrylate tube was used as a substrate collection chamber (papaya
waste). Maximum values of $0.736 \pm 0.204$ V and $5.57 \pm 0.45$ mA were generated, while pH values
increased from 3.848 to $8.227 \pm 0.35$ and Brix decreased slowly from the first day. The maximum
power density value was 878.38 mW/cm$^2$ at a current density of 7.245 A/cm$^2$ at a maximum voltage
of 1072.77 mV. The bacteria were identified with an identity percentage of 99.32% for *Achromobacter
xylosoxidans* species, 99.93% for *Acinetobacter bereziniae*, and 100.00% for *Stenotrophomonas maltophilia*.
This research gives a new way for the use of papaya waste for bioelectricity generation.

**Keywords:** waste; papaya; microbial fuel cells; generation; electricity

## 1. Introduction

Microbial fuel cells (MFCs) are electrochemical devices capable of producing bio-
electricity from wastewater, organic waste or sewage sludge, based on the conversion of
chemical energy into electrical energy [1,2]. Among the wide variety of MFCs, there are
single-chamber MFCs, which contain the anode and cathode electrodes in the same con-
tainer, but are connected by an external circuit (where the electrons flow) and most of the
time they are connected by a proton exchange membrane inside the cell [3]. MFCs generate
electricity from microbial activity present in substrates (fuels), which is done through the
oxidation of organic matter. One of their main advantages is their long functional life and
their low cost of production, in addition to the positive impact on the environment [4,5].

On the other hand, papaya is one of the most important fruits in the subtropical and
tropical regions of the world. It is known worldwide for its high content of vitamins A and
C, potassium, folic acid, niacin, thiamine, riboflavin, iron, calcium and fibre, and its high
calcium content [6]. 100 g of this fruit contains an average of 950 I.U. of vitamin A and
60.9 mg of vitamin C, as well as 38.6% ascorbic acid, 5.6% protein, 0.225% phosphoric acid,
8.3% carbohydrates, 0.0064% iron, and minerals such as magnesium (0.035%). According
to Singh et al. (2020), 80% of the world's population depends on this fruit for primary
health care [7]. Likewise, many agroindustries have dedicated their efforts to improving
desired agronomic characteristics such as size, sweetness and fruit shape for exportation

mainly to the United States due to product demand [8]. In 2018 India (with 5.99 tonnes), Brazil (with 1.06 million tonnes), and Mexico (with 1.04 tonnes) were the main producers of papaya [9]. The increase in demand was because the fruit can be consumed as fresh fruit or as a processed derivative such as ice cream, dried papaya and sweets [10]. Recently, Oladipo et al. (2020) used papaya (i.e., Carica Papaya) as a source of green catalyst for the transesterification of oil and alcohol [11].

In this context, many companies are importing and exporting this fruit, which produces large amounts of fruit waste due to the harvest or errors in the industrial process [12]. Because of this, papaya waste has great potential to be used as a substrate (fuel) in MFCs for generating electricity due to the carbohydrate content present in the decomposition of organic matter. In this sense, recently, Rahman et al. (2021) studied in single-chamber MFCs the voltage generation from mango, orange and banana waste, being the MFCs with orange waste the one that managed to generate the highest voltage (357 mV) at room temperature, followed by banana and mango waste [13]. Likewise, Ghazali et al. (2019) used grape bunch waste as a substrate in single-chamber MFCs, managing to generate approximately 0.5 V and $825 \pm 3.08 \, mW/m^2$ voltage and power density, respectively, at room temperature, also demonstrating that, for low temperatures, voltage generation decreases [14]. Kebaili et al. (2021) showed in their MFCs that electricity generation increases in aerobic systems compared to anaerobic systems. They generated maximum peaks of 230 and 140 mV for aerobic and anaerobic systems, respectively, and electroactive bacteria were found in aerobiosis [15].

This research created, low-cost microbial fuel cells were created, using papaya waste as substrate (fuel). Values of voltage, current, pH, conductivity and °Brix were monitored for 35 days. Power and current density, as well as the internal resistance of MFC, were characterised by Fourier-transform infrared spectroscopy (FTIR), and the identification of electrogenic bacteria adhered to the anode was made by using the VITEK automated system. This will provide an alternative to generate eco-friendly electricity and, at the same time, to add value to the waste of this fruit, benefiting not only society but also companies and/or farmers involved in this area.

## 2. Materials and Methods

### 2.1. Construction of Single-Chamber Microbial Fuel Cells

Microbial fuel cells (three in total) were created by using copper (Cu) electrodes in the anode and zinc (Zn) in the cathode as shown in the prototype of Figure 1, in the absence of a proton-exchange membrane. A 600 mL polymethylmethacrylate tube was used as an MFC chamber, to which a 5 cm hole was drilled at one end for the cathode to have contact with the environment (O₂), the electrodes were 78.54 cm² in the area; both electrodes were joined by an external resistor connected with copper wire (0.2 cm in diameter).

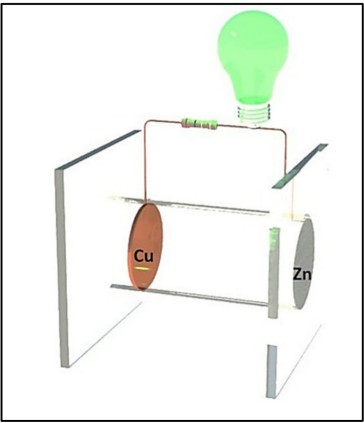

**Figure 1.** Scheme of the MFC prototype.

### 2.2. Collection and Preparation of Papaya Waste

Two kilos of decomposed papaya were collected from La Hermelinda Trujillo market, Perú, which were placed in sealed airtight bags and stored in a cooler. The wastes were placed in plastic trays to remove any remains of plastic, paper or other foreign material. This was followed by washing the samples with distilled water three times to eliminate dust, insects or other impurities. Samples were then left to dry in an oven (Labtron, LDO-B10) for 24 h at 30± °C. Lastly, papaya residues were placed in an extractor (Maqorito-400 rpm) to obtain 600 mL (150 mL for each CCM) of papaya waste juice.

### 2.3. Characterisation of Microbial Fuel Cells

The voltage and current values generated were monitored using a multimeter (Prasek Premium PR-85) for a period of 35 days with an external resistor of 1000 Ω. The current density (CD) and power density (PD) were calculated using Equations (1) and (2) (Rojas-Flores et al., 2020), where A (area) of the cathode has an approximate value of 78.5398 cm$^2$. With external resistors ($R_{ext.}$) of 0.3 ($\pm$0.1), 0.6 ($\pm$0.18), 1 ($\pm$0.3), 1.5 ($\pm$0.31), 3 ($\pm$0.6), 10 ($\pm$1.3), 20 ($\pm$6.5), 50 ($\pm$8.7), 60 ($\pm$8.2), 100 ($\pm$9.3), 120 ($\pm$9.8), 220 ($\pm$13), 240 ($\pm$15.6), 330 ($\pm$20.3), 390 ($\pm$24.5), 460 ($\pm$23.1), 531 ($\pm$26.8), 700 ($\pm$40.5), and 1000 ($\pm$50.6) Ω [16]. Changes in conductivity (conductivity meter CD-4301), pH (pH-meter 110Series Oakton) and degrees Brix (RHB-32 Brix refractometer) were also measured. Transmittance values were measured by FTIR (Thermo Scientific IS50) and resistance values of MFCs were measured using an energy sensor (Vernier- $\pm$30 V & $\pm$1000 mA).

$$PD = \frac{V_{cell}^2}{R_{ext.}A} \qquad (1)$$

$$CD = \frac{V_{cell}}{R_{ext.}A} \qquad (2)$$

### 2.4. Isolation of Electrogenic Microorganisms in Anodic Chamber

The isolation of electrogenic microorganisms inside the anodic chamber was done by replicating the experimental method from Rojas et al. 2021, [17]. Whereas, molecular recognition was done by the Analysis and Investigation Center from "Biodes Laboratorios". Deoxyribonucleic Acid (DNA) bacterial extraction was done from pure culture media by the Cetyl Trimethyl Ammonium Bromide (CTAB) [18] method. In addition, sequencing (Macrogen laboratory in the U.S) of specific polymerase chain reaction (PCR) products for the gen 16S ribosomal region of bacteria was done [19]. Sequencing analysis was done using bioinformatic software Molecular Evolutionary Genetics Analysis (MEGA) X. This was followed by the alignment and comparison with other sequences from the Basic Local Alignment Search Tool (BLAST) software, resulting in the identification percentage of the bacteria.

## 3. Results

Voltage values were increased from the first day (0.919 $\pm$ 0.002 V) until day 16 (1.029 $\pm$ 0.131 V), and, then, they decreased until the last day (0.736 $\pm$ 0.204 V) of monitoring, as it can be seen in Figure 2a. Increases and decreases in voltage values are due to the increase and depletion of nutrients in anodic behaviour [20]. The reduction in voltage values of MFCs is also because, as time passed, the organic matter content increased, depositing on the bottom of the chambers and, thus, the size of organic particles increased, which restricted the movement of protons in the substrate of the anode chamber [21]. When is because, when the organic load exceeds a specific concentration, the voltage decreases due to the increase in internal resistance [22]. This research shows better voltage values compared to other research; for example, Kondaveeti et al. (2019) used pineapple waste for power generation managing maximum voltage peaks of 275 mV in approximately 800 h (33 days) [23]. Similarly, Mbugua et al. (2020) used carbon and graphite electrodes in their cells with combined fruit and vegetable waste substrates, managing a maximum voltage

peak of approximately 135 V on day 8 [24]. Likewise, Toding et al. (2018) used orange and banana waste in their MFCs, being the cell with banana substrate the one that managed to generate approximately 0.68 V on the first day. This one is much higher than in the cell with orange waste on the first day (0.46 V) [25]. Figure 2b shows the current values generated during the 35 days of monitoring, being the tenth day the maximum current peak (5.57 ± 0.45 mA); it decreased slowly until the last day (1.66 ± 0.61 mA). According to Yang et al. (2019), the electrons released in the process of the current generation are pulled by some cellular enzymes at some points during the redox process, and these enzymes were present in the biofilm of the anode electrode, which would generate changes in the values of current generated [26]. Likewise, the rapid coupling of microorganisms with Cu anode could have helped current values to increase rapidly, but, after some time, they began to decrease. This could be because copper is a harmful material for the microorganisms in charge of generating current [27]. Although the use of metallic electrodes is a good alternative due to the high electrical conductivity of this type of material, it should be coated with a biocompatible material, so that the electrode does not have adverse effects on the microorganisms responsible for generating electrical current [28,29].

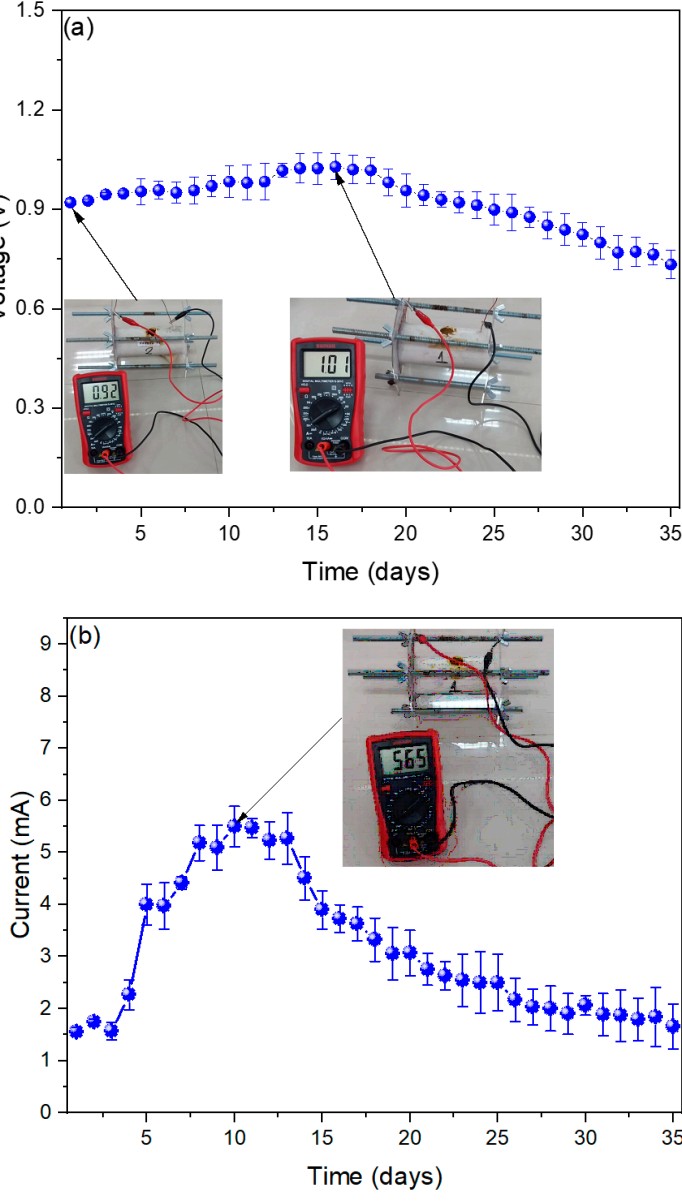

**Figure 2.** Monitoring of (**a**) voltage and (**b**) current values of MFCs during 35 days.

Figure 3a shows the average values of conductivity of MFCs monitoring for 35 days. The values increase from the first day (152.7 ± 1.01 mS/cm) to the eighth day (180.25 ± 7.9 mS/cm) and, then, they decrease until the last day (87.32 ± 4.56 mS/cm). According to Stefanova et al. [30], probably, the increase in electrical conductivity is due to reducing the internal resistance of fuel (organic waste). Likewise, in other studies, it has been clearly proven that the electrochemical parameters of MFC could be improved by the addition of inorganic salts to the substrate of the anode chamber [31]. Figure 3b shows the pH values of MFCs during monitoring. As can be seen, values increase from a moderately acidic level on the first day of (3.848) to a slightly alkaline level on the last day (8.227 ± 0.35). In the research conducted by Margaria et al. (2017), it was observed that for each substrate there is an optimum pH for the generation of electric current in MFC that for each substrate. Based on this, we could say that the optimum pH for our research was on the tenth day when it showed a pH of 6.09 ± 0.13 [32]. pH variation is mainly because protons are equimolecularly consumed with electrons in the reduction reaction [33], revalidating thus what Shanmuganathan et al. (2020) said, that there is a contradiction in the literature about the ideal pH conditions for each substrate, due to the influence of the interaction of synergistic or antagonistic of parameters [34]. While, finally, °Brix values can be observed in Figure 3c, whose values decrease from the first day until day 28, when it shows zero °Brix; then, they show no variations until day 35. It is worth mentioning that papaya contains proteins, lipids, dietary fibre, glucides, β-carotene, vitamin C, total minerals, iron and calcium [35]. These are the fuels consumed by some microorganisms for the synthesis of new cellular components.

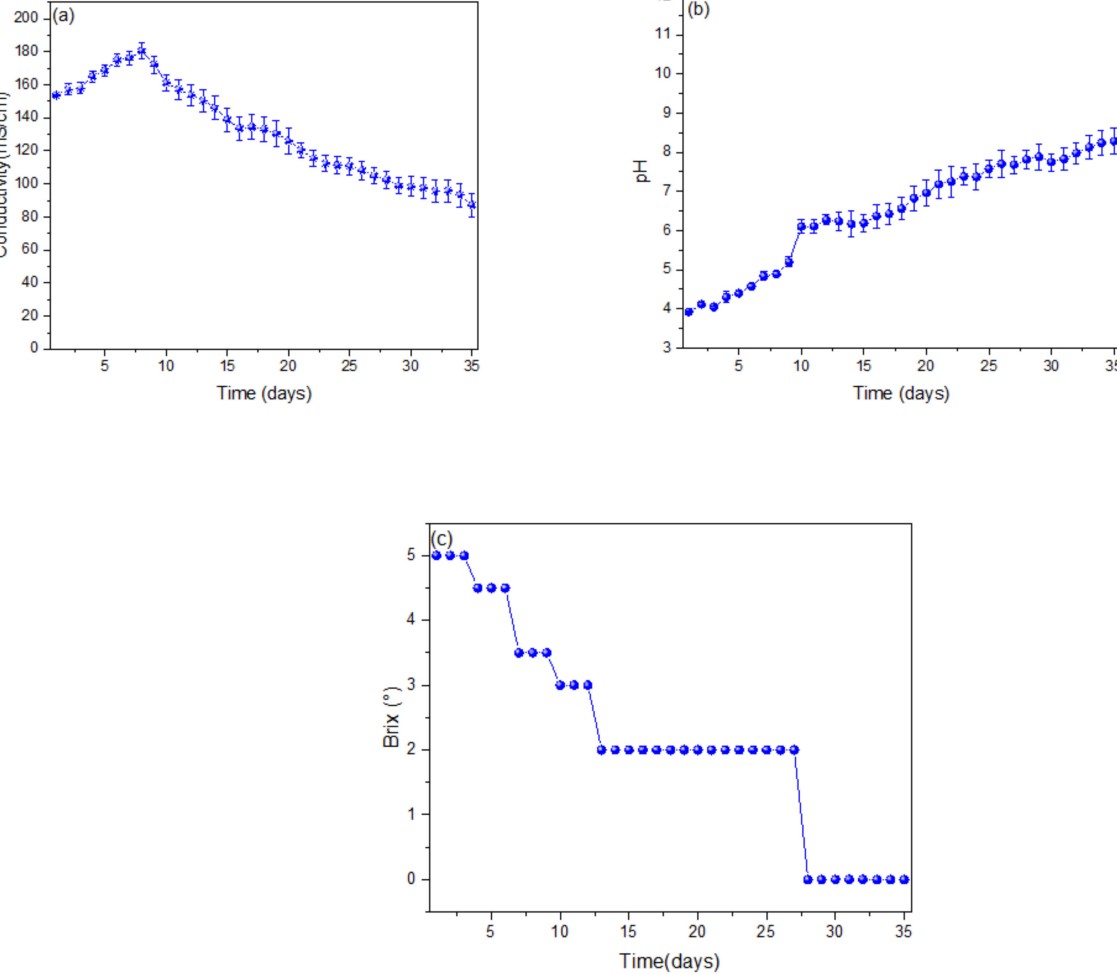

**Figure 3.** Monitoring of (**a**) conductivity, (**b**) pH and (**c**) brix parameters of MFCs.

Figure 4a shows the modelling of Ohm's Law, which can be described by multiplying the current (I) by resistance (R) to obtain the voltage (V), i.e., V = IR. Therefore, the *x*-axis was assigned to the current, while the potential was assigned to the *y*-axis. The experimental data fit the equation y = 347.7x + 0.06829 with $R^2$ = 0.555. The slope represents the average resistance of MFCs (347.7Ω); whose value is lower than that of the actual external resistor (1000 Ω) applied during the experiments. The low resistance value was due to the formation of the electroconductive biofilm on the surface of the anode and cathode [36]. Figure 4b shows the current density (CD), power density (PD) and voltage values generated from MFCs; $PD_{max}$ was 878.38 mW/cm$^2$ at a CD of 7245 A/cm$^2$ with a maximum voltage of 1072.77 mV. These values are much higher than those generated in the work of Kalagbor and Akpotayire (2020), in which the weight of fruit waste (watermelon and papaya) varied, generating a $PD_{max}$ of 1.9349 mW/cm$^2$ for the cell with 7 kg of substrate [37]. This large variation may be because the substrate was not crushed before placing it in the MFC. This is corroborated by the research of Flores et al. (2020), in which they worked with citrus extract waste, where $PD_{max}$ was 72 $\pm$ 1.4 mW/cm$^2$ for MFC with mandarin extract substrate [38]. Similarly, Ihesinachi et al. (2020) used papaya waste in extract and managed to generate $PD_{max}$ values of 2269.36 W/m$^2$. Although this value is higher than our research, it may be due to the quantity (20 kg) of substrate used and cell dimensions (for 25 L) to obtain this result [39].

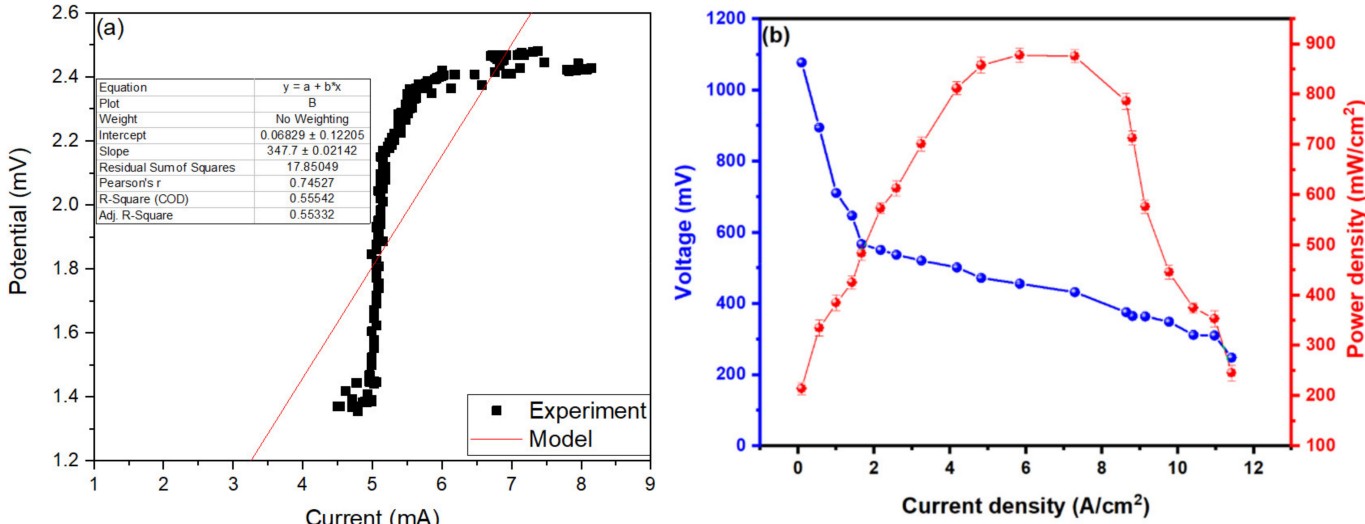

**Figure 4.** Characterisation of (**a**) internal resistance and (**b**) power and voltage density in relation to the current density of MFCs.

Figure 5 shows the FTIR spectrum to observe the changes that could have occurred due to the adhesion of the biofilm to the anode electrode and the generation of bioelectricity during the 35 days of monitoring. Thus, FTIR spectra show the initial and final spectrum of the substrate (papaya waste) in a spectral range of 4000–450 cm$^{-1}$ with a spectral resolution of 4 cm$^{-1}$. The peak at 3381 cm$^{-1}$ is due to OH and NH stretching, which can be assigned to the moisture content of the decomposing substrate. The peak of 2928–2850 is due to the presence of lipid, and the peak of 1642 cm$^{-1}$ is due to the presence of proteins and amide. Finally, the most characteristic peaks in the region 1500–500 cm$^{-1}$ are 1067, 1006 and 634 cm$^{-1}$, and are associated with the stretching vibrations of starch and the anomeric region [40–42]. The decrease in the intensity of peaks is mainly due to the breaking of bonds in the process of degradation and generation of bioelectricity [43]. According to Kale et. al. (2018) the decrease in OH bond peaks is due to the frequent breaking of the aliphatic chain resulting from chemical reactions and/or Vander Waals forces [44]. Similarly, Cholassery et al. (2019) affirm that the decrease of the bands around the peak 995 cm$^{-1}$ is due to the complete conversion of the sucrose into ethyl alcohol product of the fermentation [45]. In

the same way, the presence of the initial peaks between 900 and 1400 cm$^{-1}$ confirms the presence of glucose and fructose which, according to Sakata et al. (2020), are components that generate bioelectricity [46].

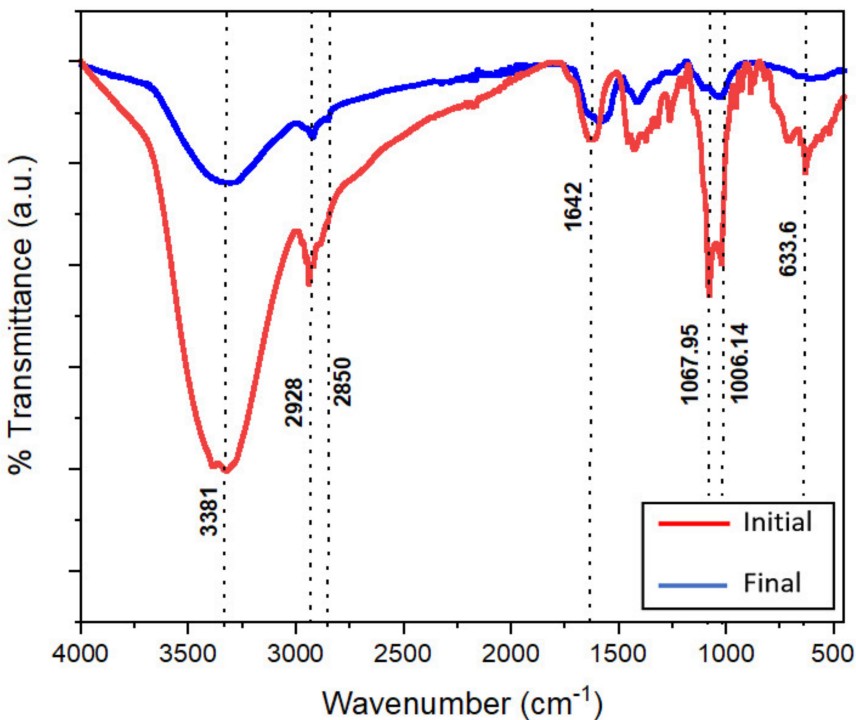

**Figure 5.** FTIR spectrophotometry of the initial and final papaya waste.

Table 1 shows the results from the BLAST characterisation of the DNA sequence from isolated bacteria at the microbial fuel cell anode. The use of the ribosomal ribonucleic acid (rRNA) 16S is the best method for bacterial identification, providing phylogenetic information from sequencing analysis [47]. By comparing the results to the database, identity percentages of 99.32% for *Achromobacter xylosoxidans*, 99.93% for *Acinetobacter bereziniae*, and 100.00% for *Stenotrophomonas maltophilia* were obtained. Xue et al. (2017) reported a biocathode's bacterial diversity of a microbial combustion cell for the Cr (VI) reduction in tannery effluents where the authors emphasise the classes of *Stenotrophomonas* sp., *Stenotrophomonas maltophilia*, *Serratia marcescens* and *Acromobacter xylosoxidans* [48].

**Table 1.** BLAST characterisation of rRNA sequence of isolated bacteria at the MFC's anode with papaya juice substrate.

| BLAST Characterisation | Sequence Length (nt) | Maximum Identity % | Identification Number | Phylogeny |
|---|---|---|---|---|
| *Achromobacter xylosoxidans* | 1451 | 99.32% | CP053617.1 | Cellular organisms; Bacteria; Proteobacteria; Betaproteobacteria; Burkholderiales; Alcaligenaceae; Achromobacter |
| *Acinetobacter bereziniae* | 1468 | 99.93% | CP018259.1 | Cellular organisms; Bacteria; Proteobacteria; Gammaproteobacteria; Pseudomonadales; Moraxellaceae; Acinetobacter |
| *Stenotrophomonas maltophilia* | 1477 | 100.00% | NR_041577.1 | Cellular organisms; Bacteria; Proteobacteria; Gammaproteobacteria; Xanthomonadales; Xanthomonadaceae; Stenotrophomonas; Stenotrophomonas maltophilia group |

Figure 6 was done using MEGA X software using the Maximum Likelihood with a thousand replications. DNA sequencing based on the rRNA 16S was employed for the detection of the bacteria. These bacteria are cosmopolitans, residing in humid environments [49]. Such bacteria are phylogenetically related with the MFC anode's isolate bacteria, grouping them by the RNA 16S analysis, and producing various phylogenetic lengths. Afta et al. (2020) found that *Stenotrophomonas* sp., *Stenotrophomonas maltophilia*, *Serratia marcescens*, *Achromobacter xylosoxidans*, Nitrobacterias and βproteobacterias [50] are useful to efficiently catalyse electron transport in microbial fuel cells [51]. In the same manner, Zafar et al. (2019) performed the molecular phylogenetic analysis (i.e., pyrosequencing) of their cells with environmental samples, revealing a general change in bacterial density and diversity. After enrichment at the second stage with oil-contaminated soil and thermal water, it was observed the greatest change in bacterial species on the anode surface such as *Stenotrophomonas maltophilia* (89%) and *Shewanella* sp. (e15%) [52]. Figure 7 shows the processing scheme for bioelectricity generation by using papaya waste. The three microbial fuel cells were connected in series generating 2.92 V on day 12, which was sufficient to light a LED bulb (blue).

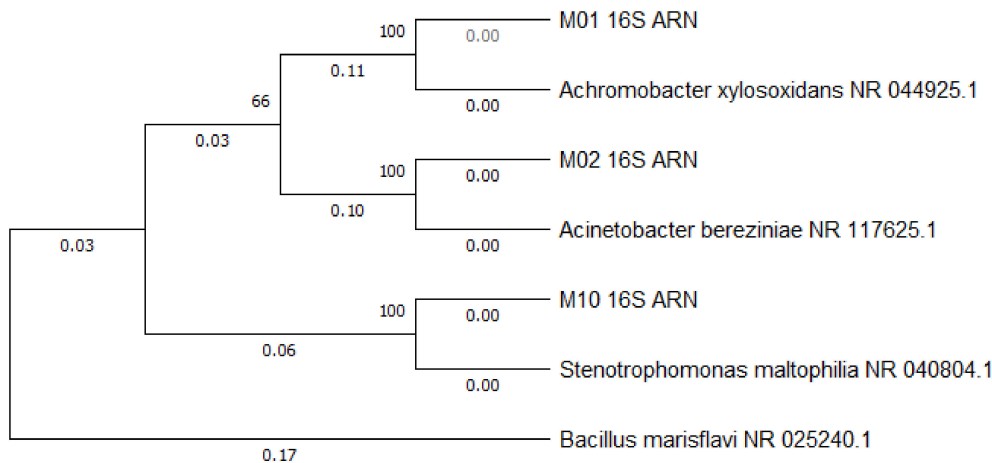

**Figure 6.** Dendrogram of bacterial groups isolated at the MFC's anode with papaya juice substrate.

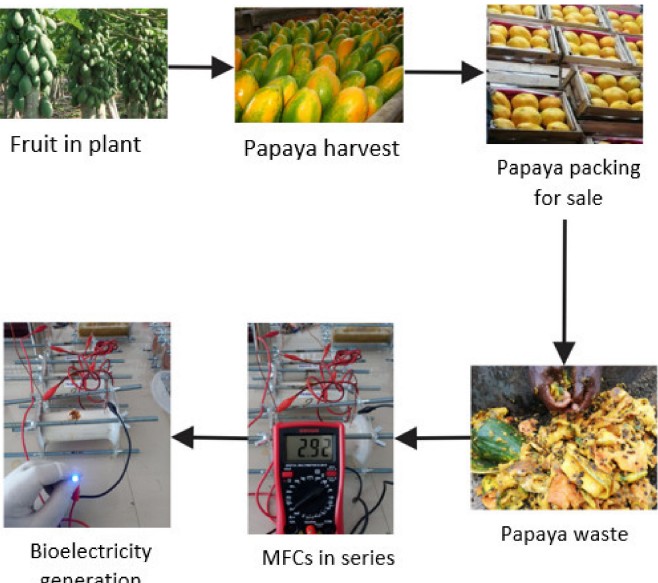

**Figure 7.** Scheme of Electricity Production by using MFCs with papaya waste in series.

## 4. Conclusions

This research successfully generated bioelectricity by using papaya (*Carica papaya*) waste through low-cost microbial fuel cells, using as a camera a tube of polymethyl-methacrylate, and electrodes of zinc and copper. Maximum voltage and current peaks of $1.029 \pm 0.131$ V and $5.57 \pm 0.45$ mA were generated on the sixteenth and tenth day, respectively. Similarly, the substrate showed conductivity peaks of $180.25 \pm 7.9$ mS/cm and showed an optimum pH of $6.09 \pm 0.13$, while °Brix decreased to zero on day 28. $PD_{max}$. was $878.38$ mW/cm$^2$ at a $CD_{max}$. $7.245$ A/cm$^2$ with a maximum voltage of $1072.77$ mV, and an internal cell resistance of $347.7$ $\Omega$. The initial and final FTIR spectra show the decrease in transmutation peaks due to bond breaking during monitoring. The different bacteria were identified using the BLAST software and an identity percentage of 99.32% for Achromobacter berreziniae and 100.00% for Stenotrophomonas maltophilia was found. Finally, cells connected in series were able to light a LED light bulb, showing encouraging results for future large-scale applications. Because of this, this research promises to provide eco-friendly solutions and to give a second use to the waste from fruit import and export industries.

**Author Contributions:** Conceptualisation, S.R.-F.; methodology S.M.B.; software, R.N.-N.; validation, H.R.-A. formal analysis, S.R.-F. and M.D.L.C.-N.; investigation S.R.-F. data curation, M.D.L.C.-N. and N.M.O.; writing—original draft preparation, O.P.-D.; writing—review and editing, S.R.-F. and O.P.-D.; project administration, S.R.-F. and R.N.-N. All authors have read and agreed to the published version of the manuscript.

**Funding:** This research received no external funding.

**Institutional Review Board Statement:** Not applicable.

**Informed Consent Statement:** Not applicable.

**Data Availability Statement:** Not applicable.

**Acknowledgments:** The authors thank Juan Pastrana, for the help provided in the grammar of the English language.

**Conflicts of Interest:** The authors declare no conflict of interest.

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
