# Peer review of "Potential Use of Papaya Waste as a Fuel for Bioelectricity Generation"

_processes, doi:10.3390/pr9101799_

Round 1
Reviewer 1 Report
Current manuscript describes the research for use of papaya wastes in microbial fuel cells. Experimental design and analysis for the main parameters of MFC, including voltage, current and pH, were performed well. However, the discussion and conclusion are bit poor compared to the well-written results. If a few points could be in-depth revisions, it may be approved for publication in the Processes journal. See the comments provided.
- I hope that the overall resolution of the figures will be higher. For example, figure1, figure 3(a) and figure 4(b) will be much easier understand for readers if the resolution is increased.
- In figure 5, initial and final graph cannot be distinguished. Graph legend should be revised.
- It was very good to identify representative bacteria, but it would be better to discuss the proportions that make up their consortium. For example, when 16S RNA was analyzed for several colonies, knowing the ratio of how many each bacteria was found, it is considered to be of great help in subsequent research.
- The last paragraph of results and conclusion section have been completely copied. It is absolutely unacceptable in the journal. Conclusion sections should be changed.
- Even though bootstrap was used in the phylogenetic analysis of figure 6, the value is too low. In general, bootstrap values should be at least higher than 70% to be meaningful. Please re-analyze this part or discuss it in-depth.
- Materials 2.4.1 and 2.4.2 do not need to be divided. Recommend to unify it to 2.4.
- There is no discussion section at all. In the research paper, there must be a discussion about the cases where there are various interpretations of the results or points to be considered more in the future. Although some discussions is included in the results section, I recommend to write a more discussion in detail.
Author Response
Dear colleague, thank you very much for the comments made to improve the manuscript.
I hope you like the improved work.
Best regards

Reviewer 2 Report
I made some sentence structure and grammatical corrections to the manuscript. These changes do not affect the overall quality of the paper but merely increases comprehension.
Figure 5 needs attention. The legend is not visible in the current state and author is requested to make appropriate edits.

Author Response

(The authors gave the same response as above.)

Reviewer 3 Report
The authors present an interesting work that can contribute to improving the management of organic waste on the planet. However, the length, wording and description of the methods used leaves much to be desired. It is necessary to make certain major corrections for publication.
The introduction is scarce and needs to be expanded. In addition, it is known that Papaya residues are used in the industry for different applications, it would also be convenient to include the different possibilities offered by this raw material as a general approach to the problem.
The methodology can also be greatly improved. It is necessary to expand it and it would be convenient to better describe the treatment of the waste, as well as to better explain how the Characterization of Microbial Fuel Cells has been carried out
The results are clear, although the discussion is scarce and needs to be improved. In addition, some images and texts in figures 2 and 4 are impossible to read / visualize.
It is recommended to use several subsections in the results and to improve the discussion of the tables. The article gets a bit lost with reading, it is interesting to focus it and highlight the results obtained.
The conclusions are very scarce and limited, they must be expanded.
Some of the articles included in the bibliography do not contain the DOI.
Author Response

(The authors gave the same response as above.)

Round 2
Reviewer 3 Report
The authors have made all the changes proposed by the reviewer.